# Allogeneic Stem Cell Transplantation in Relapsed/Refractory Multiple Myeloma Treatment: Is It Still Relevant?

**DOI:** 10.3390/jcm9082354

**Published:** 2020-07-23

**Authors:** Hyunkyung Park, Ja Min Byun, Sung-Soo Yoon, Youngil Koh, Dong-Yeop Shin, Junshik Hong, Inho Kim

**Affiliations:** 1Department of Internal Medicine, Seoul National University Boramae Medical Center, Seoul 07061, Korea; spikenice@hanmail.net; 2Department of Internal Medicine, Seoul National University College of Medicine, Seoul National University Hospital, Seoul 03080, Korea; go01@snu.ac.kr (Y.K.); stephano.dyshin@gmail.com (D.-Y.S.); alertjun@hanmail.net (J.H.); kim_dajung@hanmail.net (I.K.); 3Cancer Research Institute, Seoul National University College of Medicine, Seoul 03080, Korea; 4Center for Medical Innovation, Biomedical Research Institute, Seoul National University Hospital, Seoul 03080, Korea

**Keywords:** multiple myeloma, allogeneic stem cell transplantation, survival, high risk

## Abstract

Background: Despite offering an attractive option, the role of allogeneic stem cell transplantation (alloSCT) for treatment of multiple myeloma (MM) remains unclear. Methods: Recognizing the paucity of data in the Asian population, we retrospectively evaluated the outcomes of 24 patients (median age 52) undergoing alloSCT between April 2003 and November 2017. Results: The median time from diagnosis to alloSCT was 39.4 months. The majority of the patients (70.8%) underwent alloSCT followed by reduced intensity conditioning regimens after a median of five lines of therapy. Among 24 patients, 15 patients (62.5%) had a high-risk MM feature. The two-year relapse-free survival (RFS) and overall survival (OS) of the total patients were 29.2 ± 9.3% and 44.3 ± 10.3%, respectively. Patients who were treated with less chemotherapy lines (<5) before alloSCT had a prolonged RFS and OS. All patients (seven patients) who received a myeloablative conditioning regimen had high-risk features, but two out of seven patients showed long-term survival without lasting sequelae. Nine patients (37.5%) experienced non-relapse mortality (NRM) within one year after alloSCT (the one-year cumulative incidence of NRM was 38.3 ± 10.1%). Conclusion: AlloSCT can still be implemented as effective salvage option in the treatment of relapsed/refractory high-risk MM. The optimal timing of alloSCT remains to be determined.

## 1. Introduction

The landscape of treatment options for relapse/refractory multiple myeloma (MM) continues to evolve dramatically. The advent of new generations of proteasome inhibitors (PI) [1,2], immunomodulatory drugs (IMiD) and now immunotherapies [3,4,5,6] has considerably improved survival. Unfortunately, however, these agents do not definitely eliminate MM cells and MM remains fundamentally an incurable disease. In this context, allogeneic hematopoietic stem cell transplantation (alloSCT) remains an only potentially curative treatment option [7].

The role of alloSCT in MM treatment is a topic of much controversy, mainly due to high morbidity and mortality rates and its cost-effectiveness [8]. The general consensus is to consider alloSCT within a clinical trial setting and for highly selected patients with early relapse after primary therapy that includes an autologous hematopoietic stem cell transplantation (autoSCT) and/or those with high-risk features [9]. In reality, the use of alloSCT is based on individualized decisions and used more liberally as evident by the steadily increasing number of cases [10].

Currently, the majority of the available data on alloSCT in MM is heavily geared towards the Caucasian population [11]. Considering that the treatment choices and subsequently the outcomes of MM are particularly governed by treatment option availabilities and health policies [12], the apparent paucity of Asian data propelled us to investigate the outcomes of alloSCT in Korean MM patients. Korea has a sole public medial insurance system that covers approximately 98% of the overall population and the range of coverage is strictly uncontrolled [13]. Thus, there is the advantage of a standardized treatment algorithm consistent throughout the country. We also sought to delineate the prognostic factors for alloSCT.

## 2. Materials and Methods

### 2.1. Study Design and Subjects

This was a retrospective, longitudinal cohort study of MM patients over 18 years old treated at Seoul National University Hospital. The study period was set between January 2003 and December 2017. A total of 24 patients were identified and their medical records were reviewed for demographics, disease characteristics, treatment schema, details of alloSCT, response to treatment and survival outcomes. This study was conducted according to the Declaration of Helsinki and was approved by the Institutional Review Board of Seoul National University Hospital (IRB No.H-2001-130-1096). The informed consent was waived in light of the retrospective nature of the study and the anonymity of the subjects. 

### 2.2. Details of alloSCT

All patients received conditioning chemotherapy, which was followed by alloSCT on day 0. The conditioning regimens consisted of myeloablative (MAC) or reduced intensity (RIC) regimens. The MAC regimens included total body irradiation (TBI, daily 300 rad on days −7 to −4)-melphalan (Mel, 100 mg/m^2^ on days −3 and −2), busulfan (Bu, 3.2 mg/kg on days −6 to −3)-fludarabine (Flu, 40 mg/m^2^ on days −6 to −3) and thiotepa (200 mg/m^2^ on days −9 to −7)-Bu (2.7 mg/kg on days −6 to −4)-cyclophosphamide (Cy, 60 mg/kg on days −3 and −2). The RIC regimens consisted of Cy (60 mg/kg on days −3 and −2)-Flu (30 mg/m^2^ on days −6 to −2), Mel (90 mg/m^2^ on days −3 and −2)-Flu (30 mg/m^2^ on days −6 to −2) and Bu (3.2 mg/kg on days −7 and −6)-Flu (30 mg/m^2^ on days −7 to −2). All patients received a recombinant granulocyte colony-stimulating factor from day 1 of the stem cell transplantation until the absolute neutrophil counts (ANC) were >1000/μL for three consecutive days or >3000/μL. Patients were treated with cyclosporine (3 mg/kg) or tacrolimus (0.04 mg/kg/day) with or without a short course of methotrexate (15 mg/m^2^ on day 1 and 10 mg/m^2^ on days 3, 6 and 11) or anti-thymocyte globulin (ATG) (1.5 mg/kg/day or 2.5 mg/kg/day on days −3 to −1) or post-transplant Cy (50 mg/kg on days 3 and 4) as graft-versus-host disease (GVHD) prophylaxis. 

### 2.3. Definitions

Neutrophil engraftment was defined as an ANC > 0.5 × 10^9^/L on three consecutive measurements. Platelet recovery was defined as two consecutive measurements of 20.0 × 10^9^/L without transfusion. Acute GVHD grading was performed according to the standard criteria [14]. Chronic GVHD was classified as mild, moderate or severe according to the 2014 National Institutes of Health consensus criteria [15]. The disease status and response to treatment were evaluated according to the International Myeloma Working Group response criteria [16]. Non-relapse mortality (NRM) was defined as death without the progression of the underlying multiple myeloma. The relapse-free survival (RFS) was defined as the time from stem cell infusion to relapse or death from any cause. The overall survival (OS) was defined as the time from stem cell infusion to death of any cause. The graft-versus-host disease-free, relapse-free survival (GRFS) after alloSCT was defined as the time from stem cell infusion to grade 3-4 acute GVHD, systemic therapy-requiring chronic GVHD, relapse or death [17].

### 2.4. Statistical Analysis

The differences between groups were assessed using a student’s *t*-test or one-way analysis of variance for continuous variables and the Pearson chi-square test for categorical variables, as indicated. The RFS, OS and GRFS curves were estimated using the Kaplan-Meier method. If patients survived without death or progression, the survival was censored at the latest date of follow-up when no death or progression was confirmed. The Kaplan–Meier method was also used to display the cumulative incidence of acute and chronic GVHD, relapse and NRM. 

Clinical variables with *p* values of <0.05 in the univariate analyses were included in the multivariate analyses performed using the Cox proportional hazard model. All statistical tests were two sided and the significance was defined as a *p* value of <0.05. All analyses were performed with SPSS version 22.0 (IBM, Armonk, NY, USA).

## 3. Results

### 3.1. Patient Characteristics

The baseline clinical characteristics of all 24 patients are summarized in Table 1. The median age at the time of alloSCT was 52 years old (range, 37–65 years old). The cell sources of all cases were peripheral blood stem cells as per national policy. There were 15 patients with high-risk features [18]: six with ISS stage III, six with plasmacytoma, two with p53 deletion and one with t (4;14). Plasma cell leukemia patients were subjected to a median of five lines of treatment including autoSCT and the median time from MM diagnosis to alloSCT was 39.4 months (range, 5–121.4 months). There were four patients who did not receive autoSCT prior to alloSCT: two due to rapid progression and two with high-risk cytogenetics per attending physician’s choice. The prior two were subjected to salvage alloSCT while the latter two were subjected to upfront alloSCT. The majority of the patients received RIC regimens (17/24 patients, 70.8%) from full-matched related donors. Their disease was at least partially controlled at the time of alloSCT (partial response or better in 13/24 patients, 54.2%). There were seven patients undergoing a MAC regimen. For these patients, MAC was chosen in light of their high-risk features including plasmacytoma in four and presence of p53 deletion in two.

### 3.2. The alloSCT Outcomes

As shown in Table 2, the overall response rate (ORR: complete remission [CR] + very good partial response [VGPR] + PR) was 62.5% (15/24). Twenty-one patients achieved neutrophil engraftment at a median of day 13. The three patients who did not achieve neutrophil engraftment died due to NRM within 100 days after alloSCT. Eighteen patients achieved platelet engraftment at a median of day 17.5. Among the six patients who did not achieve platelet engraftment, five patients died due to NRM and one patient survived without engraftment. Eleven patients experienced relapse after alloSCT and, among them, seven patients received further chemotherapy. None were subjected to additional hematopoietic stem cell transplantation. 

During the median follow-up of 10.8 months (range, 0.5–73.5 months), the two-year RFS, OS and one-year GRFS ± standard errors (SE) for all the patients were 29.2 ± 9.3%, 44.3 ± 10.3% and 25 ± 8.8%, respectively (Figure 1A for RFS and Figure 1B for OS). The median RFS, OS and GRFS were 4.9 (95% confidence interval [CI], 0–9.9), 22.3 (95% CI, 3.2–19.1) and 2.2 months (95% CI 0.1–4.2) months, respectively. In the univariate and multivariate analyses (Table 3), treatment with less than five chemotherapy lines before alloSCT was associated with longer RFS (2-year RFS: 66.7 ± 19.2% for <5 lines versus 16.7 ± 8.8% for ≥5 lines; *p =* 0.006; Figure 1C) and a trend towards a longer OS (2-year OS: 66.7 ± 19.2 for <5 lines versus 36.4 ± 11.7 for ≥5 lines; *p =* 0.137; 1D). Additionally, a lower ISS stage was related to a prolonged RFS and OS (Table 3).

### 3.3. The Complications of alloSCT

Acute GVHD ≥ grade 2 occurred in 29.2% of the patients (7/24) (Table 2). The cumulative incidence of acute GVHD for 100 days was 30.6% (Figure 2A). Apart from one patient who experienced grade 4 skin GVHD, all cases of acute GVHD were well-controlled. The cumulative incidence of chronic GVHD for one year was 13.7% (Figure 2B). Notably, chronic lung GVHD ≥ grade 2 occurred in two patients: one patient undergoing alloSCT from a haploidentical donor using a RIC Bu-Flu regimen and cyclosporine, methotrexate and ATG as GVHD prophylaxis; the other undergoing alloSCT from a full-matched unrelated donor using Thio-Bu-Cy and cyclosporine and post-transplant Cy as GVHD prophylaxis. The prior patient expired due to combined pneumonia while the latter patient survived to undergo lung transplantation. 

The one-year cumulative incidence of NRM for the total patients was 38.3% (Figure 2C); five patients died because of sepsis, two patients died from intracranial hemorrhage, one patient died from pneumocystis carinii pneumonia and cytomegalovirus infection and one patient died from grade 4 acute skin GVHD and infection. The intensity of the conditioning regimen did not affect the one-year cumulative incidence of NRM (14.3% for the MAC regimen versus 47.1% for the RIC regimen, *p* = 0.386).

### 3.4. Long Term Survivors

At the time of the data collection there were five patients who were alive and the median OS after alloSCT for these five patients was 3.6 years (Table 4). Details of these patients are shown in Table 4. Among them, three patients relapsed after a median 22.8 months of alloSCT and underwent further chemotherapy. Among the two patients who did not experience a relapse after alloSCT, one underwent upfront alloSCT with a TBI-based MAC regimen in light of the aggressiveness of her disease. She experienced acute skin GVHD (grade 3) after the alloSCT but it resolved after treatment with steroid, tacrolimus and ruxolitinib. The other patient obtained CR after alloSCT with RIC conditioning. She experienced acute liver GVHD (grade 2), which was controlled with steroid treatment.

## 4. Discussion

We conducted this study in the hope of establishing the role of alloSCT in MM treatment and found (1) alloSCT can be an attractive salvage option with tolerable rates of GVHD; (2) earlier implementation of alloSCT may lead to better outcomes and (3) in patients with high-risk features, alloSCT with MAC regimens may bring long-term survival. Most patients had been treated with new available agents, such as IMiD and PI, before alloSCT. Therefore, this study may be meaningful because it emphasizes the role of alloSCT in patients who have resistance to these new agents.

The present study showed that the two-year RFS and OS for the total patients were 29.2 ± 9.3% and 44.3 ± 10.3%, respectively. In particular, two patients (8%) who had high-risk features were cured after alloSCT (Table 4). Our results were comparable with those reported in previous studies despite the difference in the composition of patients, with patients in our cohort being more heavily treated prior to alloSCT [19,20,21] (Table 5). Heavily treated patients have limited treatment options due to the heterogenous drug resistance mechanism of MM cells [22,23]. Therefore, considering most patients were heavily treated and 62.5% (15/24) of patients had high-risk features (median OS: 2–3 years in the previous report) [24], our study may verify the role of alloSCT in the era of highly potent agents. On the other hand, the rate of NRM seemed to be higher in our group per the reported one-year NRM by El-Cheikh et al., which is 17% [21]. This difference might be due to the disease status of our patient population. In our data, among the nine patients who experienced NRM within one year, eight patients were treated with at least five lines of chemotherapy and four patients had stable disease or progressive disease status before alloSCT. These findings that chemo-resistant patients experienced more treatment-related mortality and lower RFS were already shown in previous reports [25,26]. The current study also showed that heavily pretreated patients showed significantly worse RFS and OS (Figure 1C,D).

As for the timing of the alloSCT, recent studies suggest that upfront alloSCT in newly diagnosed MM showed a significantly improved RFS and OS compared with the relapsed setting [27,28]. Similarly, a Swedish group reported multicenter prospective data that auto/alloSCT groups showed better survival outcomes and a lower relapse rate compared with autoSCT-only in previously untreated MM patients [29]. Although a direct comparison is difficult as most of our patients underwent alloSCT in relapsed/refractory settings, there was one patient with a high risk myeloma who underwent upfront alloSCT after induction and remained disease free at post-alloSCT year 3.4 (Table 4). In NRM, early alloSCT may also have lower toxicity; in the case of NRM within one year in this study, only one patient (1/9 patients) who was treated with 1–4 lines experienced NRM. However, considering that early intervention has relatively less toxicity, it is insufficient that upfront alloSCT could be associated with lower toxicity compared with standard therapy. For conformation of this, a further randomized, well-controlled trial is needed. In a previous multicenter autoSCT study performed in Korea (KMM 103 study), 3/41 patients (7.3%) experienced mortality within one year due to sepsis in patients who were treated with autoSCT following bortezomib-based induction chemotherapy [30]. Although this study showed relatively lower NRM compared with our study, considering that this study only included newly diagnosed MM patients, it may be a comparable outcome. Based on our experience and cumulated data, we can tentatively surmise that alloSCT might be a good curative option in high-risk subgroups such as high-risk cytogenetics, extramedullary disease and plasma cell leukemia [9,31]. However, since alloSCT is a challenging procedure [32], careful selection of patients is the key to successful treatment. Furthermore, we have to take into consideration that while most previous studies assess disease outcomes for endpoints, only a few studies report the quality of life or their deterioration after alloSCT. In this study, we estimate the quality of life using GRFS. However, there may be a limitation for evaluation of quality of life using this estimate. Therefore, a comprehensive approach is necessary for future studies.

It is the general consensus that a RIC regimen is preferable [33] and this trend was also seen in our cohort. Previous comparative data of the two conditioning regimens showed that RIC had a lower NRM, but the relapse rate was higher compared with MAC regimens [11]. Long term follow-up research conducted by Sahebi et al. reported that 10% of patients who received RIC-alloSCT experienced late relapse after 6–12 years post-transplant [34]. We could not effectively compare between the two conditioning regimens because most patients (70.8%) received RIC conditioning. However, although all patients who were treated with a MAC regimen had high-risk features, 2/7 patients (28.6%) showed long-term survival (Table 4). Additionally, a MAC regimen showed no significant difference in NRM compared with a RIC regimen. Therefore, a MAC regimen could be a reasonable option for selected patients with high-risk features. However, interpretation requires caution because of the small sample size. 

This study has several limitations. First, it is a retrospective study with a small number of patients. Second, cytogenetic data at diagnosis was missing in 45.8% of the patients. This might be because our data collected from the year 2003 was during the non-computerized era and some patients transferred from other hospitals during the treatment. Third, many patients were not treated with novel agents including carfilzomib or daratumumab prior to alloSCT. Nevertheless, this study adds to our understanding of the role of alloSCT in MM treatment, especially in the Asian population. 

In conclusion, alloSCT can be a beneficial treatment option for a selected group of MM patients even in heavily treated settings. Patients with high-risk myelomas seem to benefit the most from the procedure. 

## Figures and Tables

**Figure 1 jcm-09-02354-f001:**
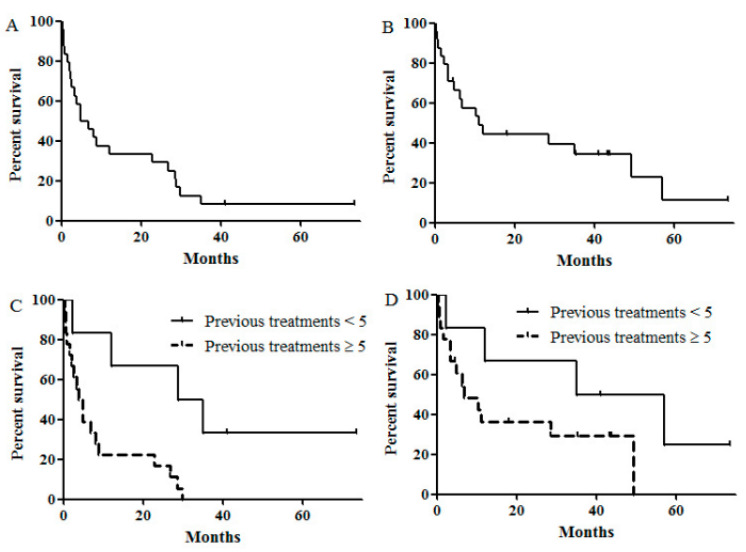
Survival outcomes. (**A**) Relapse free survival (RFS) all patients (2-year RFS: 29.2 ± 9.3%) (**B**) Overall survival (OS) of all patients (2-year OS: 44.3 ± 10.3%) (**C**) RFS according to the previous treatment of chemotherapy lines (2-year RFS: 66.7 ± 19.2% for <5 lines versus. 16.7 ± 8.8% for ≥5 lines, *p* = 0.006) (**D**) OS according to the previous treatment of chemotherapy lines (2-year OS: 66.7 ± 19.2% for <5 lines versus 36.4 ± 11.7% for ≥5 lines, *p* = 0.137).

**Figure 2 jcm-09-02354-f002:**
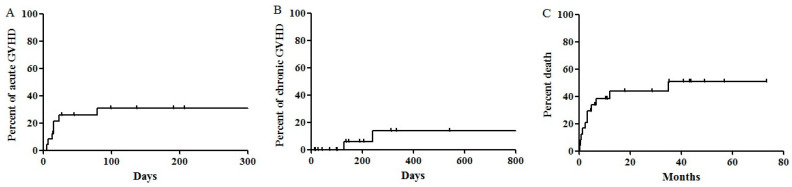
Allogeneic stem cell transplantation outcomes. (**A**) Acute graft-versus-host disease (GVHD) (cumulative incidence of acute GVHD for 100 days: 30.6 ± 9.7%) (**B**) Chronic GVHD (1-year cumulative incidence of chronic GVHD: 13.7 ± 9.2%) (**C**) Non-relapse mortality (one-year cumulative incidence of NRM: 38.3 ± 10.1%).

**Table 1 jcm-09-02354-t001:** Baseline characteristics of patients. MM, multiple myeloma; ISS, International Staging System; Ig, immunoglobulin; AutoSCT, autologous stem cell transplantation; alloSCT, allogeneic stem cell transplantation; CR, complete response; VGPR, very good partial response; PR, partial response; SD, stable disease; PD, progressive disease; HLA, human leukocyte antigen; MAC, myeloablative conditioning; RIC, reduced intensity conditioning; GVHD, graft-versus-host disease; MTX, methotrexate; M, male; F, female. * High risk cytogenetics: p53 deletion for one patient, t (4;14) for one patient and p53 deletion + t (14;16) for one patient.

Patient Characteristics	Patients (*n* = 24)
Median age, years (range)	52 (37–65)
Sex, *n* (%) Male/Female	15 (62.5)/9 (37.5)
MM with plasmacytoma/plasma cell leukemia	8 (33.3)
Durie Salmon stage at diagnosis	
1/2/3	2 (8.3)/5 (20.8)/10 (41.7)
Unknown	7 (29.2)
ISS stage at diagnosis	
1/2/3	4 (16.7)/7 (29.2)/6 (25.0)
Unknown	7 (29.2)
Type of light chains	
Kappa/Lambda	8 (33.3)/13 (54.2)
Unknown	3 (12.5)
Isotype of M-protein	
IgG/IgA/IgM	11 (45.8)/2 (8.3)/0
IgD/light chain	0/4 (16.7)
Unknown	7 (29.2)
Cytogenetics	
High risk */Standard risk	3 (12.5)/10 (41.7)
Unknown	11 (45.8)
Median previous treatment lines, numbers (range)	5 (1–9)
Previous treatment	
Bortezomib-based treatment/refractoriness	21 (87.5)/14/21 (66.7)
Thalidomide-based treatment/refractoriness	21 (87.5)/14/21 (66.7)
Lenalidomide-based treatment/refractoriness	6 (25.0)/4/6 (66.7)
One autoSCT/Two autoSCT/refractoriness	18 (75.0)/2 (8.3)/20/20 (100.0)
The information of alloSCT	
Pre-alloSCT status	
CR/VGPR/PR	3 (12.5)/3 (12.5)/7 (29.2)
SD/PD	8 (33.3)/3 (12.5)
Donors	
Sibling/Matched-unrelated/Haploidentical	17 (70.8)/6 (25.0)/1 (4.2)
HLA matching	
Full matching	20 (83.3)
9/10/8/10/4/8	1 (4.2)/2 (8.3)/1 (4.2)
Conditioning regimens	
MAC regimen	7 (29.2)
RIC regimens	17 (70.8)
GVHD prophylaxis	
Cyclosporine/Tacrolimus/MTX	21 (87.5)/3 (12.5)/4 (16.7)
Anti-thymocyte globulin	12 (50)
Post-cyclophosphamide	1 (4.2)
Donor-Recipient sex	
M-M/F-F/M-F/F-M	9 (37.5)/4 (16.7)/6 (25.0)/5 (20.8)
Median infused cells (CD34+) (range)	4.58 (1.77–28.68) × 10^6^/kg
Median levels of M protein before alloSCT, (range)	0.45 (0–6.9)
Median time from diagnosis to alloSCT, months (range)	39.4 (5.0–130.0)

**Table 2 jcm-09-02354-t002:** The outcomes after alloSCT.

Outcomes after alloSCT	Patients (*n* = 24)
Median follow-up periods after alloSCT, months (range)	10.8 (0.5–73.5)
Best response after alloSCT	
CR/VGPR/PR	10 (41.7)/1 (4.2)/4 (16.7)
SD/PD	6 (25.0)/0
Unknown	3 (12.5)
Median time to neutrophil engraftment, days (range)	13 (9–23)
Median time to platelet engraftment	17.5 (13–90)
Relapse after alloSCT	
Cumulative incidence of relapse (one year)	62.5% (±9.9)
Further treatment after alloSCT	
Yes	7 (29.2)
GVHD after alloSCT	
Acute GVHD (≥ Grade 2)	7 (29.2)
Cumulative incidence of acute GVHD (100 days)	30.6% (±9.7)
Skin/GI tract/Liver	3 (12.5)/1 (4.2)/4 (16.7)
Chronic GVHD (≥ Grade 2)	2 (8.3)
Cumulative incidence of chronic GVHD (1 year)	13.7% (±9.2)
Lung	2 (8.3)
Non-relapse mortality (within one year)	9 (37.5)
Cause of death	
Sepsis	5 (20.8)
Intracranial hemorrhage	2 (8.3)
Acute GVHD and infections	1 (4.2)
PCP and CMV infection	1 (4.2)

AlloSCT, allogeneic stem cell transplantation; CR, complete response; VGPR, very good partial response; PR, partial response; SD, stable disease; PD, progressive disease; GVHD, graft-versus-host disease; GI, gastrointestinal; PCP, pneumocystis carinii pneumonia; CMV, cytomegalovirus.

**Table 3 jcm-09-02354-t003:** The univariate and multivariate analyses for relapse-free survival and overall survival.

		Univariate Analysis		Multivariate Analysis			Univariate Analysis		Multivariate Analysis		
Variables		2-Year RFS (%)	*p* Value	HR	95% CI	*p* value	2-Year OS (%)	*p* Value	HR	95% CI	*p* Value
Age, years	≥52	25.0 (±12.5)	0.686				47.6 (±15.0)	0.636			
	<52	33.3 (±13.6)					41.7 (±14.2)				
Periods	2003–2009	25.0 (±15.3)	0.195				37.5 (±17.1)	0.202			
	2010–2017	31.3 (±11.6)					47.7 (±12.9)				
Durie Salmon stage	1	50.0 (±35.4)	0.351				50.0 (±35.4)	0.513			
	2	80.0 (±17.9)					80.0 (±17.9)				
	3	10.0 (±9.5)					30.0 (±14.5)				
ISS stage	1	75.0 (±21.7)	0.007	1		0.041	75.0 (±21.7)	0.005	1		0.058
	2	28.6 (±17.1)		2.125	0.462–9.767	0.333	57.1 (±18.7)		1.210	0.125–11.707	0.869
	3	0		10.238	1.513–69.278	0.017	0		6.433	0.683–60.735	0.104
High-risk myeloma [18]	High-risk	20.0 (±10.3)	0.879				36.7 (±12.9)	0.878			
	None	44.4 (±16.6)					55.6 (±16.6)				
Cytogenetics	High	66.7 (±27.2)	0.232				66.7 (±27.2)	0.970			
	Standard	20.0 (±12.6)					46.7 (±16.6)				
HLA matching	Full match	25.9 (±9.7)	0.320				42.9 (±14.4)	0.947			
	Mismatch	50.0 (±25.0)					50.0 (±25.0)				
Conditioning regimens	MAC	14.3 (±13.2)	0.880				34.3 (±19.5)	0.807			
	RIC	35.3 (±11.6)					47.1 (±12.1)				
Previous treatment lines	≥5	16.7 (±8.8)	0.006	3.035	0.772–11.932	0.112	36.4 (±11.7)	0.137			
	<5	66.7 (±19.2)		1			66.7 (±19.2)				
Pre-alloSCT status	CR	66.7 (±27.2)	0.130				50.0 (±35.4)	0.026	1	0–1.963E278	0.969
	Non-CR	23.8 (±9.3)					35.9 (±10.8)		272,589.913		
Infused cells (CD34+)	≥4.58	33.3 (±13.6)	0.145				50.0 (±14.4)	0.182			
	<4.58	25.0 (±12.5)					38.9 (±14.7)				
Pre-alloSCT M-protein	≥0.45	25.0 (±12.5)	0.220				38.1 (±14.7)	0.101			
	<0.45	33.3 (±13.6)					50.0 (±14.4)				

HR, hazard ratio; CI, confidence interval; RFS, relapse-free survival; OS, overall survival; ISS, International Staging System; HLA, human leukocyte antigen; MAC, myeloablative conditioning; RIC, reduced intensity conditioning; alloSCT, allogeneic stem cell transplantation; CR, complete response.

**Table 4 jcm-09-02354-t004:** Long term survival patients.

Age	ISS Stage	Previous Treatments	Cytogenetics	Pre-alloSCT Status	Donors/HLA Matching	Conditioning Regimens	Infused CD34 +	Chimerism after alloSCT	Relapse/RFS	Treatment after alloSCT	OS after alloSCT(Years)	OS (Years)
54	2	TD#3►VD#5►AutoSCT►LD#16►DCEP#3	IgH rearrangement/Plasmacytoma	PR	Sibling/8/8	Bu-Flu (MAC)	4.66	Complete	Relapse/22.8 months	PomD#15►KD#1	3.6	6.7
61	1	TD#3►VD#6►AutoSCT►LD#7►PomD#4	N/A	SD	Sibling/8/8	Bu-Flu (RIC)	3.38	Complete	Relapse/29.9 months	Daratumumab#14►KD#1	3.7	9.0
37	2	VTD#5	IgH rearrangement, Rb1 deletion. P53 deletion /Plasmacytoma	PR	Unrelated/ 10/10	TBI (300 rad, 4 days)/Mel (MAC)	5.88	Complete	Non-relapse/41.0 months	None	3.4	4.0
57	1	TD#4►autoSCT►VD#9	Normal cytogenetics /Plasmacytoma	CR	Unrelated/8/10	Bu-Flu (RIC)	28.68	Complete	Non-relapse/73.5 months	None	6.1	9.7
65	2	PAD#3►autoSCT►CTD#4►autoSCT►VD#3	Normal cytogenetics	VGPR	Sibling/6/6	Bu-Flu (RIC)	7.94	Complete	Relapse/ 8.2 months	VD#5►LD#6►PomD#4►bendamustine#7	3.0	6.2

Ig, immunoglobulin; ISS, International Staging System; alloSCT, allogeneic stem cell transplantation; HLA, human leukocyte antigen; RFS, relapse-free survival; OS, overall survival; TD, thalidomide + dexamethasone; VD, bortezomib + dexamethasone; autoSCT, autologous stem cell transplantation; LD, lenalidomide + dexamethasone; DCEP, dexamethasone + cyclophosphamide + etoposide + cisplatin; PomD, pomalidomide + dexamethasone; VTD, bortezomib + thalidomide + dexamethasone; PAD, bortezomib + doxorubicin + dexamethasone; CTD, cyclophosphamide + thalidomide + dexamethasone; CR, complete response; VGPR, very good partial response; PR, partial response; SD, stable disease; BuFlu, busulfan + fludarabine; RIC, reduced intensity conditioning; MAC, myeloablative conditioning; TBI, total body irradiation; Mel, melphalan; KD, carfilzomib + dexamethasone.

**Table 5 jcm-09-02354-t005:** The comparison with previous studies.

Reference	Pawarode A, et al. [19]	El-Cheikh J, et al. [21]	Our Study
Country	USA	France	Korea
Number of patients	22 patients with high-risk or advanced refractory MM	Total 53 patients/22 patients (42%) with higher-risk disease	Total 24 patients/15 patients with high-risk feature (62.5%)
Median previous treatment lines	2 (1–4)	–	5 (1–9)
NRM/relapse rate	One-year NRM: 19%/37% at one year	One-year NRM: 17%/–	One-year NRM: 38.3%/62.5% at one year
RFS/OS	Three-year RFS: 15%/three-year OS: 29%	10-year RFS: 24%/10-year OS: 32%	Two-year RFS: 29.2%/two-year OS: 44.3%
Cumulative incidence of acute GVHD/chronic GVHD	23% at day 180 (grade 3–4)/68% at one year	38% at two-year (grade 2–4)/59% at two years	30.6% at day 100/13.7% at one year
Comments	Prospective study/Using MA regimen but reduced-toxicity regimen, consisting of fludarabine and busulfan	RIC regimens/long-term outcomes (minimum follow-up of five years)	Using MAC and RIC regimens

MM, multiple myeloma; NRM, non-relapse mortality; RFS, relapse-free survival; OS, overall survival; GVHD, graft-versus-host disease; RIC, reduced intensity conditioning; MAC, myeloablative conditioning.

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
