# Peer review of "Allogeneic Stem Cell Transplantation in Relapsed/Refractory Multiple Myeloma Treatment: Is It Still Relevant?"

_jcm, 2020, doi:10.3390/jcm9082354_

Round 1

Reviewer 1 Report

I applaud the authors for making the important effort to retrospectively assess outcomes after alloSCT for patients with multiple myeloma in the Korean population.  By doing so, they make a significant contribution to the existing data on this therapeutic modality for this disease.  Although the n of 24 is likely too low to detect any differences in outcome that may exist between the various subgroups analyzed (e.g. cytogenetic risk, HLA match/mismatch, etc.), the authors do their best to analyze all of the relevant patient- disease- and treatment-related factors. 

I do have questions/issues regarding statistical values reported in the results section and conclusions drawn in the discussion section:

1) Table 2 states that non-relapse mortality (NRM) occurred within 1 year in 9(25.0) patients. What is the 25.0? Assuming this is the percentage of total alloSCT patients, 9/24=0.375 or 37.5%. So cumulative incidence of NRM at 1 year is neither 25% nor 38.3% (the stated cumulative incidence of NRM at one year given in Figure 2), it is 37.5%. Please clarify the calculations used in Table 2 and Figure 2.

2) Table 2 also lists a low “1-year relapse rate” of 29.2%, using the total patient population (n=24) as the denominator. However, 9/24 patients died from NRM within one year of transplant, so we have no idea whether these 9 patients would have relapsed or not within one year. Therefore, the “1-year relapse rate” as calculated is a meaningless and potentially misleading statistic. A more meaningful value to report would be the proportion of patients evaluable for relapse at 1 year who had in fact relapsed, aka: (everyone who relapsed within 1 year(n=7)/(total patients(n=24) minus patients who experienced NRM at 1 year(n=9)) = 7/15 = 46.6%, which is to say that almost half of the cohort who did not die of infection, brain hemorrhage, or GVHD within 1 year, suffered a relapse of their MM within 1 year.  Suggest either reporting this value or arguing in the discussion section why "relapse rate/1-year relapse rate" as currently defined are meaningful endpoints to report independent of RFS.

3) Why do the 1-year cumulative incidence rates in Figure 2 have standard deviations? These are not median or mean values, they are actual numbers of patients over a specified time period. Please explain the statistical analyses used to generate these numbers.

4) In the discussion section, the authors make the conclusion from their findings that “we can tentatively surmise that alloSCT might be a good curative option in high-risk subgroups, such as high-risk cytogenetics, extramedullary disease, and plasma cell leukemia”.  Respectfully, I fail to see how this conclusion regarding cure can be drawn from these data.  If the authors are to make this argument, then in my opinion the percentage of patients in the entire cohort who remained alive after alloSCT with even the possibility of being cured (2/24 = 8%) needs to be specified.  Additionally, there needs to be a discussion as to why we should be targeting cure with alloSCT in an era of highly potent IMiDs, PIs, and CD38 mAbs, when over a third of the cohort died within a year from complications of transplant and only one patient (4%) has lived beyond 4 years post-transplant without relapse (the other "long-term" non-relapser is only 3.4 years out from alloSCT, had only one prior line of therapy, and received an IMiD-PI-dex combination regimen (current standard of care) and so it remains purely speculative whether this patient may be cured versus just enjoying a fairly standard DOR after triplet induction and myeloablative consolidation).

5) The statement that “earlier implementation of alloSCT leads to better outcomes” is too general and could perhaps be misinterpreted as a suggestion that early alloSCT itself improves the long-term outcome of MM patients, as opposed to simply stating that post-transplant survival is better for less heavily pre-treated patients than more heavily pre-treated patients. While it is true that survival within the cohort was better after alloSCT in those patients with <5 prior lines of therapy compared to ≥5 lines, that is true for any therapeutic modality. In other words, MM earlier in the course of the disease has a better prognosis than MM later in the course of the disease, regardless of the intervention. In order to make a strong argument that early alloSCT actually improves outcomes in the Korean MM population, a randomized controlled trial of early alloSCT versus standard therapy would need to be performed, or at the very least a retrospective analysis comparing survival rates in the less heavily pre-treated cohort to matched controls who were treated with standard non-alloSCT therapies.  Additionally, if the authors wish to make the argument that early alloSCT should be encouraged in the current era, then we need to know how many of the NRM events in this cohort (to include 1-year NRM events) occurred in patients who were transplanted after only 1-4 prior lines of therapy, and there are also needs to be discussion regarding the incidence of fatal toxicities from newer MM therapies that are either currently available in Korea or may soon become available, and why the authors feel that there is still a role for early alloSCT in the context of these newer therapies.

Reviewer 2 Report

This is well written paper, showing that survival outcomes of allo-HSCT for myeloma were still unfavorable, as well as their risk factors.

  1. Despite the small number of cases, the current cohort mainly consisted of patients resistant to newly available drugs, which may be advantageous of this paper. This point may be emphasized a little more.
  2. Is it possible that allo-HSCT had a negative impact on quality of life, or promoted frailty? Without information of QOL, showing GRFS may deduce it to some extent.
